

# Biodegradation of Crystal Violet dye by bacteria isolated from textile industry effluents

Dipankar Chandra Roy[1], Sudhangshu Kumar Biswas[2,3], Ananda Kumar Saha[4], Biswanath Sikdar[5], Mizanur Rahman[3], Apurba Kumar Roy[5], Zakaria Hossain Prodhan[2,6] and Swee-Seong Tang[2]

[1] Biomedical and Toxicological Research Institute, Bangladesh Council of Scientific and Industrial Research, Dhaka, Bangladesh
[2] Division of Microbiology, Institute of Biological Sciences, Faculty of Science, University of Malaya, Kuala Lampur, Malaysia
[3] Department of Biotechnology and Genetic Engineering, Faculty of Applied Science and Technology, Islamic University Kushtia, Kushtia, Bangladesh
[4] Department of Zoology, Faculty of Life and Earth Sciences, University of Rajshahi, Rajshahi, Bangladesh
[5] Department of Genetic Engineering and Biotechnology, Faculty of Life and Earth Sciences, University of Rajshahi, Rajshahi, Bangladesh
[6] Institute of Crop Science, College of Agriculture and Biotechnology, Zhejiang University, Hangzhou, China

Corresponding author
Sudhangshu Kumar Biswas,
shu_genetics@yahoo.com

## ABSTRACT

Industrial effluent containing textile dyes is regarded as a major environmental concern in the present world. Crystal Violet is one of the vital textile dyes of the triphenylmethane group; it is widely used in textile industry and known for its mutagenic and mitotic poisoning nature. Bioremediation, especially through bacteria, is becoming an emerging and important sector in effluent treatment. This study aimed to isolate and identify Crystal Violet degrading bacteria from industrial effluents with potential use in bioremediation. The decolorizing activity of the bacteria was measured using a photo electric colorimeter after aerobic incubation in different time intervals of the isolates. Environmental parameters such as pH, temperature, initial dye concentration and inoculum size were optimized using mineral salt medium containing different concentration of Crystal Violet dye. Complete decolorizing efficiency was observed in a mineral salt medium containing up to 150 mg/l of Crystal Violet dye by 10% (v/v) inoculums of *Enterobacter* sp. CV–S1 tested under 72 h of shaking incubation at temperature 35 °C and pH 6.5. Newly identified bacteria *Enterobacter* sp. CV–S1, confirmed by 16S ribosomal RNA sequencing, was found as a potential bioremediation biocatalyst in the aerobic degradation/de-colorization of Crystal Violet dye. The efficiency of degrading triphenylmethane dye by this isolate, minus the supply of extra carbon or nitrogen sources in the media, highlights the significance of larger-scale treatment of textile effluent.

## INTRODUCTION

The textile industry plays a vital role in the global economy as well as in our daily life, and is concurrently becoming one of the main sources of environmental pollution in the world in terms of quality and quantity (*Mondal, Baksi & Bose, 2017*). The textile industry consumes a larger volume of water, in which almost ninety percent appears as wastewater (*Verma, Dash & Bhunia, 2012*). Textile wastewater contains the different type of dyes as the major pollutant which is not only recalcitrant but also imparts intense color to the waste effluent (*Mondal, Baksi & Bose, 2017*). Inappropriate disposal of textile wastewater causes serious environmental problems that affect the aquatic organism adversely (*Subhathra et al., 2013*; *Mondal, Baksi & Bose, 2017*). Improper effluent disposal in aqueous ecosystems leads to reduction of sunlight penetration which in turn diminishes photosynthetic activity, resulting in acute toxic effects on the aquatic flora/fauna and dissolved oxygen concentration (*Muhd Julkapli, Bagheri & Hamid, 2014*).

The wastewater produced from the textile, dye and dyestuff industries is a complex combination of various inorganic and organic materials (*Fulekar, Wadgaonkar & Singh, 2013*). Dyes commonly have a synthetic origin and complex aromatic molecular structures which make them more stable and more difficult to biodegrade. The textile industries consume the largest amount of dyestuffs, at nearly 60–70% (*Bhattacharya et al., 2018*; *Mudhoo & Beekaroo, 2011*; *Rawat, Mishra & Sharma, 2016*) which play a vital role in preparing raw materials to pretreatment materials together with dyeing and finishing of textile materials (*Jana, Roy & Mondal, 2015*; *Sriram, Reetha & Saranraj, 2013*). Due to the wide range of dyes, fibers, process auxiliaries and final products during the dyeing processes, an ample amount (about 10–90%) of dye-stuffs that do not bind to the fibers were released into the sewage treatment system or the environmental water (*Abadulla et al., 2000*; *Zollinger, 2003*). Dye wastes represent one of the most awkward groups of pollutants because they easily may recognize by naked eyes and are non-biodegradable (*Mojsov et al., 2016*).

Triphenylmethane dyes are synthetic compounds widely used in various industries and their removal from effluents are tough, due to their higher degree of structural complexity (*Morales-Álvarez et al., 2018*). The presence of complex mixture in textile effluent directly indicates the water has been polluted, and this highly colored effluent is forthrightly responsible for polluting the receiving water (*Rajamohan & Rajasimman, 2012*).

As a result, inappropriate textile dye effluent disposal in aqueous ecosystems leads adverse impact on chemical oxygen demand (COD) and high biological oxygen demand (BOD). Their metabolites lead to toxic, carcinogenic and mutagenic effect to flora and fauna which eventually cause severe environmental problems worldwide (*Mittal, Kurup & Gupta, 2005*; *Sharma et al., 2009*).

Due to their toxic, mutagenic and carcinogenic properties as well as their contribution to the de-coloration of natural waters, the release of dyes and their metabolites into the environment is a source of concern (*Khadijah, Lee & Faiz, 2009*). Thus, precise attention should be taken into consideration on the utilization of dyes industrially. Inadequate methods have been reported for decolorizing textile effluents economically. For the

removal of synthetic dyes from the water bodies, a number of physicochemical methods, such as filtration, specific coagulation, use of activated carbon and chemical flocculation, have been used (*Olukanni, Osuntoki & Gbenle, 2006*; *Verma, Dash & Bhunia, 2012*). Using these expensive physiochemical methods, vast amounts of sludge are produced, which result in a secondary level of land pollution (*Shah, 2013*). For this reason, there is an urgent need for inexpensive and eco-friendly removal techniques of the polluting dyes. As a potential alternative, biological processes including several taxonomic groups of microbes such as bacteria, fungi, yeast together with algae have been received growing interest due to their cost-effectiveness, their production of less sludge, and their eco-friendly nature (*Kalyani et al., 2009*). Bacteria from different trophic groups can achieve a higher degree of dye-degradation and can process a complete mineralization of dyes under optimal conditions (*Asad et al., 2007*). Recently, microbial degradation of textile effluent has been reported as more economical and eco-friendly than physiochemical methods (*Shah, 2013*).

The present study aimed to isolate and characterize Crystal Violet degrading bacteria from textile industry effluents for potential use in the industrial bioremediation process.

## MATERIALS AND METHODS

### Sample collection

The untreated water and sludge samples of textile effluent were collected from two local thread dyeing plants namely Rana Textile and Bulbul Textile Industries Ltd from Kumarkhali, Kushtia, Bangladesh. Four samples, named as water-1, water-2, sludge-1 and sludge-2, were collected from stagnant textile effluents from drainage canal. The color, pH and temperature of the samples were measured and recorded. The samples were collected in sterile plastic bottles, brought to the laboratory and kept at 4 °C in refrigerator for preservation within 24 h of sampling.

### Bacterial isolation

All four samples (untreated textile effluents) were used to isolate dye decolorizing bacteria by modified enrichment culture techniques as stated by *Shah (2013)*. Steps involved enrichment, isolation and screening of dye decolorizing bacteria were: (i) 1 ml of each sample was first diluted with 9 ml of sterilized water and the stock was kept in static condition for few minutes for precipitation; (ii) 1 ml supernatant from each diluted sample was transferred into 9 ml enrichment medium and a required amount of crystal violet dye solution was added into the stock to adjust the concentration; (iii) the species showing remarkable decolorization within 24 to 72 h were streaked on 2% enrichment agar medium containing 100 mg/l of crystal violet dye; (iv) Colonies of bacteria those exhibited a clear decolorization zone around them on enrichment agar medium were picked and cultured; (v) an individual colony was then reintroduced into 9 ml enrichment medium containing Crystal Violet dye and was incubated overnight; (vi) 10% of overnight cultured isolates were inoculated into 10 ml MS medium supplemented with 1ml/l TE solution and 100 mg/l crystal violet dye and incubated overnight; (vii) 2 ml of the sample was then removed aseptically and centrifuged for 10 min at 10,000 rpm; (viii) this supernatant was used to determine the decolorization percentage of the added dye; (ix) isolates exhibited most

decolorizing efficiency were selected and preserved (in nutrient agar up to one month and in 50% glycerol up to six months) for further studies.

## Bacterial growth determination

In order to determine the effect of pH on bacterial growth, the isolated bacteria CV-S1 was cultured in nutrient broth. A twenty four hours observation was done at 35 °C temperate using 10 ml MS medium containing 10% (v/v) inoculums and 50 mg/l Crystal Violet dye of varying pH (6.00, 6.50, 7.00, 7.50, 8.00 and 8.50) . To determine the optimum temperature, degradation assay was performed from 30 to 40 °C temperature using same stock condition at pH 6.50 (*Shah, 2013*; *Prasad & Rao, 2013*).

## DNA extraction and quality analysis

The genomic DNA extraction was performed using modified CTAB method as described by *Winnepenninckx, Backeljau & Dewachter (1993)* and the quality of DNA was analyzed through Gel electrophoresis in 1% agarose gel.

## 16S ribosomal RNA sequencing for bacterial identification

Partial sequence of 16S ribosomal RNA was carried by using Applied Biosystem 3130. The bacteria-specific forward primer F27 (AGAGTTTGATCCTGGCTCAG) and reverse primer R1391 (GACGGGCGGTGTGTRCA) were used to amplify DNA fragments in PCR. The recipe of a total of 25 µl of reaction mixture was ddH$_2$O 14.75 µl, MgCl$_2$ (25 mM) 2 µl, buffer (10×) 2.5 µl, dNTPs (10 mM) 0.5 µl, Taq DNA Polymerase (5u/µl) 0.25 µl, DNA template 1 µl, forward primer (10 µM) 2 µl and reverse primer (10 µM) 2 µl. The PCR amplification was performed by Swift™ Minipro Thermal Cycler (Model: SWT-MIP-0.2-2, Singapore) using the following program: Denaturation for 5 min at 95 °C, followed by 40 cycles of 40 s of denaturation at the same temperature, annealing for 60 s at 65 °C and elongation at 72 °C for the first 2 min and followed by a final extension for 10 min. The sequence generated from the automated sequencing of PCR amplified 16S ribosomal RNA was analyzed through the NCBI BLAST (http://www.ncbi.nlm.nih.gov) program to ascertain the possibility of a similar organism through alignment of homologous sequences and the required corresponding sequences that were downloaded. The evolutionary history was inferred using the Neighbor-joining method which was performed on the Phylogeny.fr platform through online software: Muscle (v3.7), Gblocks (v0.91b), PhyML (v3.0 aLRT) and TreeDyn (v198.3) (*Dereeper et al., 2010*; *Edgar, 2004*).

## Environmental parameters optimization

Optimization of various environmental parameters (pH, temperature, initial dye concentration and inoculum size) for decolorization of Crystal Violet dye were done with some modifications of *Shah (2013)* and *Prasad & Rao (2013)*. The mixture was inoculated with the 24 h incubated bacterial culture and uninoculated crystal violet dye solutions were kept as control. Each experiment was performed in triplicate and the mean values were recorded. To detect the effect of initial dye concentration, media of different dye concentrations 50 mg/l to 200 mg/l were used while 8, 9 and 10% (v/v) of 24 h incubated inoculums were inoculated for dye decolorization.

## Assay of dye degradation/decolorization

The rate of decolorization expressed as a percentage was determined by observing the reduction of absorbance at absorption maxima (λ max). The uninoculated MS medium supplemented with respective dye was used as a reference. A total of 2 ml of reaction mixture was kept at different time intervals, and the samples were centrifuged at 10,000 rpm for 10 min to separate biomass. The concentration of dye was determined by absorbance at 660 nm. The measurement of absorbance was done by the a photo-electric colorimeter (AE-11M; Hangzhou Chincan Trading Co., Ltd, Hangzhou, China). The color removal efficiency was stated as the percentage ratio based on the following equation (*Chen et al., 2003*):

$$\text{Dye Degradation}(\%) = \frac{\text{Initial OD} - \text{Final OD}}{\text{Initial OD}} \times 100.$$

# RESULTS AND DISCUSSION

## Physical characterization of textile effluent

The observation of physical characters of the collected textile effluent samples had revealed a high load of pollution indicators. The effluent colors of three samples were black due to a mixture of different chemicals and dyes and the rest was turquoise blue due to the fact that only turquoise dye was used in the dyeing process. The pH of the tested samples was slightly acidic to neutral. Temperature of the collected sample were around 18 °C due to winter season. Physical characters of textile effluent may vary due to the mixing of different types of organic and inorganic compounds derived from different environmental conditions. Chikkara and Rana had observed the colour and smell of textile effluent sample which was black and pungent respectively at pH 9.4 (*Chhikara & Rana, 2013*) whereas Verma and Sarma tested textile waste-water which was brownish-black in color with unpleasant odor at pH 8.3 (*Varma & Sharma, 2011*).

## Isolation, screening and identification of dye degrading bacteria

On the basis of the decolorizing capacity and colony characters 3 isolates were selected from sludge-2 and the isolates were named as CV–S1, CV–S2 and CV–S3 after 72 h of incubation CV–S1 yielded up to 81.25% Crystal Violet dye degradation while the rest two CV–S2 and CV–S3, exhibited up to 64.58% and 25% dye degradation respectively (Table 1). Thus CV–S1 isolate was selected for identification.

The best sequenced portion of 580 bp of 16S rDNA of amplified 1,500 bp exhibited the highest identity (99%) with *Enterobacter* sp. HSL69 according to isolation source. The downloaded corresponding aligned sequences, shown in Table 2, revealed that the phylogenetic relationship between the isolated bacterial strains with other related bacterial strains. During phylogenetic tree construction, strain CV–S1 had formed a new branch and the homology indicated that the strain CV–S1 is under the genus *Enterobacter*. Therefore, the isolate was identified and named as *Enterobacter* sp. CV–S1. The newly formed branch confirms that the identified *Enterobacter* sp. CV–S1 is a novel species of *Enterobacter* genus (Fig. 1). Numerous potential dye decolorizing bacteria have been reported by scientists

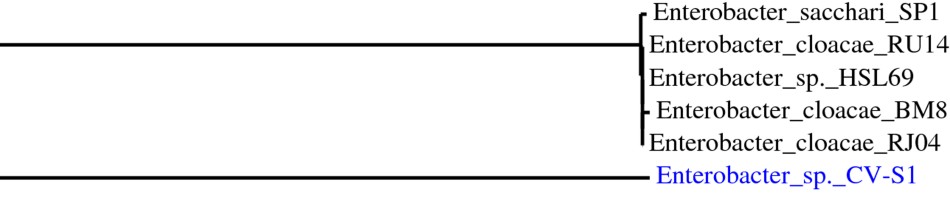

Enterobacter_sacchari_SP1
Enterobacter_cloacae_RU14
Enterobacter_sp._HSL69
Enterobacter_cloacae_BM8
Enterobacter_cloacae_RJ04
Enterobacter_sp._CV-S1

0.2

**Figure 1** **Highlighted bacterial strains are the isolated bacteria.** The phylogenetic tree was reconstructed using the maximum likelihood method implemented in the PhyML program (v3.0 aLRT) (*Dereeper et al., 2010*; *Edgar, 2004*).

**Table 1** **Screening of the best dye decolorizing isolates based on degradation rate.**

| Isolates | Initial OD | Final OD | Degradation rate (%) | Average degradation rate (%) | Duration of observation |
|---|---|---|---|---|---|
| | 0.08 | 0.015 | 81.25 | | |
| CV–S1 | 0.08 | 0.015 | 81.25 | 81.25 | 72 h |
| | 0.08 | 0.015 | 81.25 | | |
| | 0.08 | 0.03 | 62.50 | | |
| CV–S2 | 0.08 | 0.03 | 62.50 | 64.58 | 72 h |
| | 0.08 | 0.025 | 68.75 | | |
| | 0.08 | 0.06 | 25.00 | | |
| CV–S3 | 0.08 | 0.06 | 25.00 | 25.00 | 72 h |
| | 0.08 | 0.06 | 25.00 | | |

**Table 2** **Similarity between the isolated bacterial strain CV–S1 and other related bacteria found in the GenBank database.**

| Isolated strain | Closed bacteria | Accession no. | Identity (%) |
|---|---|---|---|
| | *Enterobacter cloacae* RU14 | KJ607595.1 | 99 |
| | *Enterobacter cloacae* RJ04 | KC990807.1 | 99 |
| CV–S1 | *Enterobacter cloacae* BM8 | JX514423.1 | 99 |
| | *Enterobacter* sp. HSL69 | HM461195.1 | 99 |
| | *Enterobacter sacchari* SP1 | NR_118333.1 | 98 |

from the textile dye effluents, contaminated soil with dyes, dying waste disposal sites, and wastewater treatment plant (*Khadijah, Lee & Faiz, 2009*; *Pokharia & Ahluwalia, 2013*).

## Growth characteristics

The maximum growth of CV–S1 was observed at temperature 35 °C and pH 6.50 while the growth started decreasing within 60–72 h of incubation (Table 3). Bacterial growth is a complex process associated with various anabolic and catabolic reactions. Eventually, these biosynthetic reactions result in cell division (*Raina & Charles, 2009*). As the growth-rate hypothesis (GRH) predicts positive correlations among RNA content, phosphorus (P)

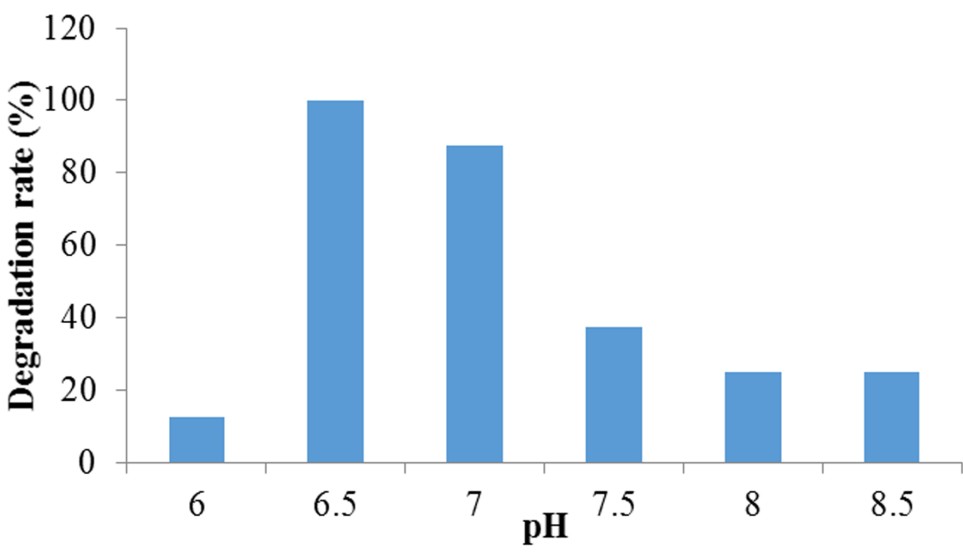

Figure 2 The effect of pH on crystal violet dye degradation by Enterobacter sp. CV–S1.

Table 3 Absorption spectra of Crystal Violet at different time intervals.

| Conc. of CV dye | Measurements | Elapsed time (in hours) | | | | | | | | |
|---|---|---|---|---|---|---|---|---|---|---|
| | | 0 | 2 | 12 | 24 | 30 | 36 | 48 | 60 | 72 |
| 150 mg/L | OD | 0.12 | 0.12 | 0.11 | 0.08 | 0.06 | 0.04 | 0.02 | 0.01 | 0 |
| | Degradation rate (%) | 0 | 0 | 8.33 | 33.33 | 50.00 | 66.67 | 83.33 | 91.67 | 100 |

content and biomass, such relationships have been used to assume patterns of microbial activity, resource availability, and nutrient recycling in ecosystems (*Franklin et al., 2011*). Hence, the degradation study required considerable of 72 h of cultivation time.

## Influence of environmental parameters on crystal violet dye degradation

The results of degradation experiment of crystal violet dye by *Enterobacter* sp. CV–S1 was involved with the effect of pH, temperature, initial dye concentration and inoculum size under aerobic shaking condition at 120 rpm.

### Effect of pH on dye degradation

This experiment revealed that the percentage of Crystal Violet dye degradation had improved with the change of pH in the medium. The higher degradation was observed at pH 6.50 to 7.00 while the highest decolorization rate (100%) was observed at pH 6.50 and lowest (12.5%) was at pH 6.00. However, organism showed very low decolorization above pH 7.50 (Fig. 2). According to the growth curve study, it was observed that the growth rate of the bacteria was higher at pH 6.5 which probably played a vital role for higher degradation at this pH level. These observations indicated that the organism could treat efficiently neutral to weakly acidic dyeing waste. Several researches had proved that
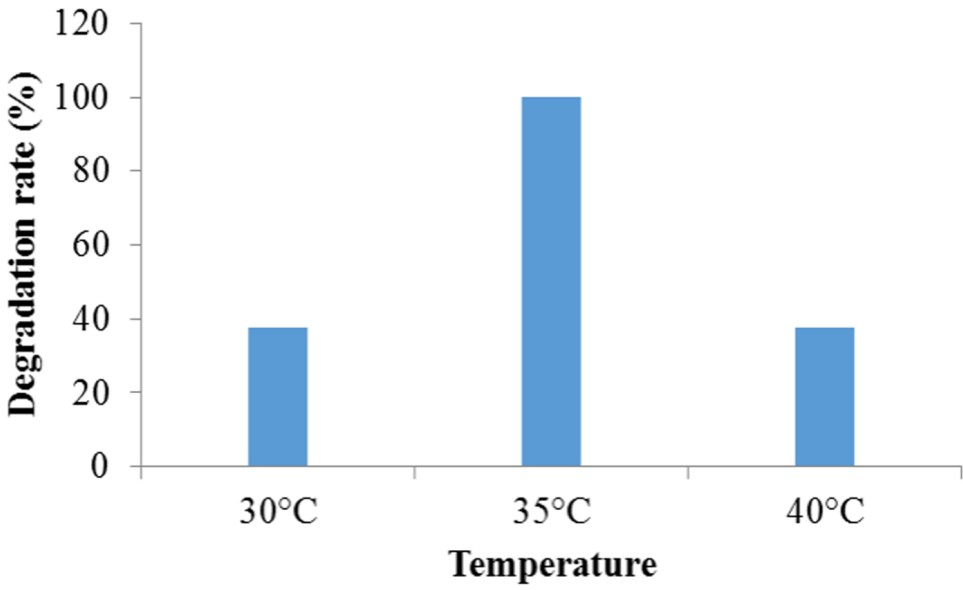

**Figure 3** **The effect of temperature on crystal violet dye degradation by Enterobacter sp. CV–S1.**

the biosorption processes using microbes were highly pH dependent (*Aksu & Tezer, 2005*; *Kumar, Ramamurthi & Sivanesan, 2006*). In another research done by *Wang et al. (2009)*, *Citrobacter* sp. CK3 had achieved the best decolorization of reactive red 180 (96%) at pH 6.0–7.0. In the case of red azo dye decoloration by *Aspergillus niger*, it was observed that the removal percent increased with the rise of pH and the maximum removal efficiency was reached (99.69%) at pH 9.0. Thereafter, whenever the pH value increases, the decolorization process appeared to decrease (*Mahmoud et al., 2017*).

### Effect of temperature on dye degradation
The maximum (100%) degradation was observed at temperature 35 °C while at temperature 30 °C and 40 °C, the much adverse effect on the degradation was found and it was 37.5% in both cases (Fig. 3). This might have occurred due to an adverse effect of lower and higher temperature other than 35 °C on the enzymatic activities and the rate of chemical reaction directly related to temperature change. In addition, bacteria needs optimum temperature for growth. Since dye decolorization is a metabolic process, the change in temperature causes change from optimum results into a decline dye decolorization. The higher temperature causes thermal inactivation of proteins and probably affects cell structures such as the membrane (*Shah, 2013*). Similar effect of temperature was observed by *Bacillus subtilis* in crystal violet dye degradation (*Kochher & Kumar, 2011*). *Holey (2015)* reported that bacterial consortium showed 98% decolorization at 100 mg/L concentration of Congo Red dye at temperature 37 °C. *Lalnunhlimi & Krishnaswamy (2016)* reported that the microbial community exhibits the optimal degradation efficiency with a temperature ranges from 30 to 35 °C. *Wanyonyi et al. (2017)* observed the optimal temperature for decolorization of Malachite Green by using novel enzyme from *Bacillus cereus* strain KM201428 at 40 °C.

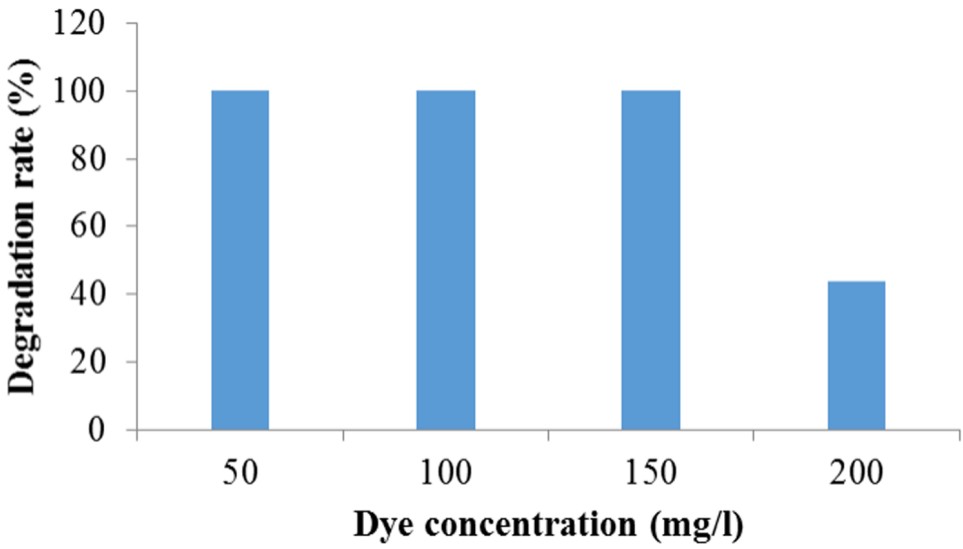

**Figure 4** Degradation of different concentration of crystal violet dye by Enterobacter sp. CV–S1.

### Effect of initial dye concentration on dye degradation

It was observed that *Enterobacter* sp. CV–S1 can degrade 150 mg/l Crystal Violet dye within 72 h. However in higher concentration, dye degradation rate was remarkably reduced (Fig. 4). This may be due to the decreasing of nucleic acids content ratio, i.e., RNA/DNA, which results to lowering the protein synthesis that inhibits cell division. The decolorization of 500 mg/l crystal violet using *Bacillus* sp. was complete upon continued incubation for 2.5 h in mineral salt medium amended with glucose and yeast extract but it decreased to less than with increasing the initial concentration of crystal violet to 600, 700, 800, 900 and 1,000 mg/l (*Ayed et al., 2009*). The effect of dye concentration on growth plays an important role in the choice of microbes to be used in the bioremediation of dye wastewater, for instance high concentrations can reduce the degradation efficiency due to the toxic effect of the dyes (*Khehra et al., 2006*). Furthermore, initial dye concentration provides an essential driving force to overcome all mass transfer resistance of the dye between the solid and aqueous phases (*Parshetti et al., 2006*). Present result indicate that *Enterobacter* sp. CV–S1 is quite tolerant to Crystal Violet and can decolorize and degrade relatively higher concentration of the dye.

### Effect of initial inoculums size on dye degradation

It was observed that the dye removal capacity was affected by the inoculums size used. The degradation rate had decreased with the declining of inoculum sizes. The most significant result (100%), was obtained when 10% inoculum was used. The absorption spectra of crystal violet at different time intervals were shown in Table 3 and Fig. 5. After 72 h of inoculation the solution was streaked on a nutrient agar plate and the growth of bacteria was observed after overnight incubation. It proved that the dye degradation was absolutely due to bacteria. After optimizing the environmental parameters, 100% degradation of

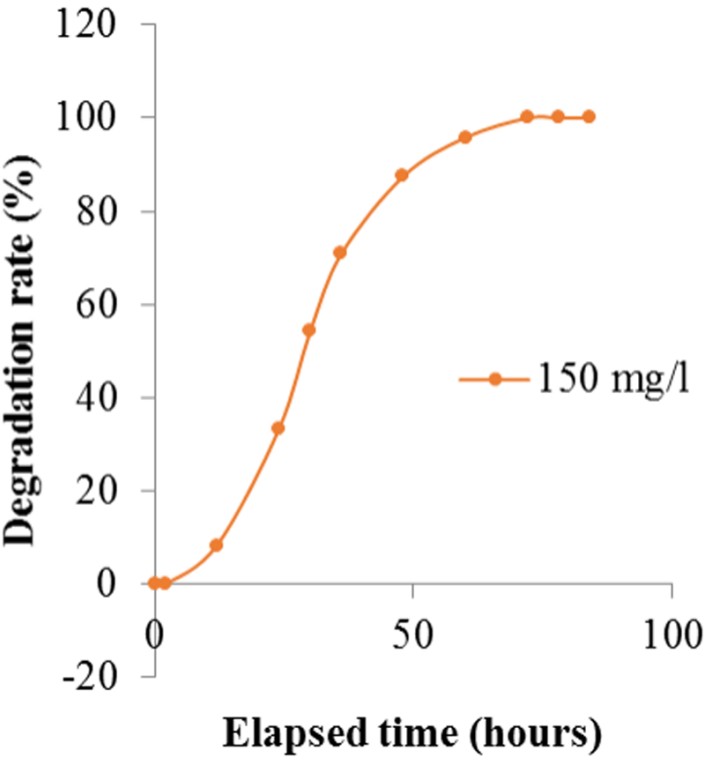

**Figure 5** Degradation rate of Crystal Violet by Enterobacter sp. CV–S1 after optimizing the environmental parameters at different time intervals.

150 mg/l Crystal Violet was observed within 72 h at 35 °C and pH 6.50 under aerobic shaking condition by 10% (v/v) *Enterobacter* sp. CV–S1 without supplying extra carbon and nitrogen source as shown in Fig. 6. A similar pattern was observed and reported by *Ayed et al. (2009)* that the dye removal capacity had increased significantly with the escalation in inoculum size. They isolated *Bacillus* sp. which was able to decolorize 500 ppm crystal violet within 2.5 h under shaking condition at temperature 30 °C and pH 7. In another study, the Brilliant Green dye (10 mg/l) removal by the *Klebsiella* strain Bz4 in static conditions was observed 81.14% after 24 h of incubation and 100% dye removal was observed after 96 h of incubation (*Zabłocka-Godlewska, Przystaś & Grabińska-Sota, 2015*).

## CONCLUSION

In this study, the newly isolated bacteria *Enterobacter* sp. CV–S1 has demonstrated potentiality for its Crystal Violet dye degradation. The optimum decolorizing parameters of the study were concentration of dye (150 mg/l), inoculums size (10% v/v) temperature (35 °C), pH (6.50), with a rotation of 120 rpm. It can be concluded from the overall findings that the isolated bacteria *Enterobacter* sp. CVS1 could effectively be used as an alternative to the physical and chemical processes of textile effluents as they have a high potential for being able to decolorize or degrade Crystal Violet dye.

**Figure 6** **10% (v/v) of *Enterobacter*. sp. CV–S1 showed 150 mg/l Crystal violet dye degradation at pH 6.50 and temperature 35 °C under shaking condition at different time intervals.** (A) 0 h; (B) 24 h; (C) 48 h and (D) 72 h (c, control; R1, R2 and R3, replication 1, 2 and 3 respectively).

# ACKNOWLEDGEMENTS

We are especially grateful to the Centre for Advanced Research in Science (CARS), University of Dhaka, Bangladesh.

## Funding

This work was funded by the Dept. of Zoology and the Dept. of Genetic Engineering and Biotechnology, University of Rajshahi, Bangladesh. The funders had no role in study design, data collection and analysis, decision to publish, or preparation of the manuscript.

## Grant Disclosures

The following grant information was disclosed by the authors:
Dept. of Zoology.
Dept. of Genetic Engineering and Biotechnology, University of Rajshahi, Bangladesh.

## Competing Interests

The authors declare there are no competing interests.

## Author Contributions

- Dipankar Chandra Roy conceived and designed the experiments, performed the experiments, analyzed the data, prepared figures and/or tables.
- Sudhangshu Kumar Biswas conceived and designed the experiments, performed the experiments, analyzed the data, contributed reagents/materials/analysis tools, prepared figures and/or tables, authored or reviewed drafts of the paper, approved the final draft.
- Ananda Kumar Saha and Biswanath Sikdar conceived and designed the experiments, contributed reagents/materials/analysis tools, authored or reviewed drafts of the paper.
- Mizanur Rahman analyzed the data, contributed reagents/materials/analysis tools, authored or reviewed drafts of the paper.
- Apurba Kumar Roy contributed reagents/materials/analysis tools, authored or reviewed drafts of the paper.

- Zakaria Hossain Prodhan analyzed the data, contributed reagents/materials/analysis tools, authored or reviewed drafts of the paper, approved the final draft, depositing sequence in gene bank for accession no.
- Swee-Seong Tang authored or reviewed drafts of the paper, approved the final draft.

## Data Availability

The raw data are provided in the Supplemental Files.

## Supplemental Information

Supplemental information for this article can be found online at http://dx.doi.org/10.7717/peerj.5015#supplemental-information.

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
