# Peer review of "Biodegradation of Crystal Violet dye by bacteria isolated from textile industry effluents"

_PeerJ, doi:10.7717/peerj.5015_

## Round 0.1 · original submission · Major Revisions

Please carefully respond to the comments from the three reviewers. Note: Reviewer 3 has attached an annotated PDF

Reviewer 1 ·

Basic reporting

In this manuscript, the authors proposed the concept of utilizing bacterial to degrade organic dyes/pollutants (crystal violet as a sample). They have optimized the conditions of pH, temperature, and reaction time. Overall, this is an interesting paper and could has impact in the near future. Some points need to be considered before the acceptance of this manuscript.
1. I strongly suggest the authors provide the absorption spectra of crystal violet at different degradation times. This will help the readers to understand more clearly about the degradation level at a specific reaction time.
2. Also, I suggest them to provide the photograph of bacteria at different degradation times. If the bacteria maintain their original color (for example, white), this means that the de-colorization of crystal violet solution is truly from the degradation by bacteria. Otherwise, the de-colorization of crystal violet solution could also happen when the crystal violet is adsorbed on bacterial.
3. Because the degradation time is 72 hours, the authors also need to prove that after such a reaction time with crystal violet, the bacteria are still alive. This will help the readers to fully believe that the degradation of crystal violet is from bacteria themselves.

Experimental design

No Comment

Validity of the findings

No comment

Reviewer 2 ·

Basic reporting

Introduction part need complete some degradation data of references with this study.

Experimental design

It is presents well and a smart design.

Validity of the findings

A good ideal for degradation of dye with CV-S1.

Additional comments

1. Why the degradation rate is highest at pH 6.5?
2. What is the lifetime of bateria CV-S1?

·

Basic reporting

1. The title should be reworded. The following two titles have been suggested:
i. Biodegradation of crystal violet dyes by bacteria isolated from textile industry effluents.
ii. Characterisation of crystal violet degrading bacteria from textile industry effluents.
2. Comprehension of the introduction section of this manuscript is difficult without thorough editing and recasting of several statements. Some areas have been highlighted in the manuscript and suggestions made.
3. Line 114-118: The objective should be recast ‘the present study aimed to isolate and characterise crystal violet degrading bacteria from textile industry effluents for potential use in industrial bioremediation process’.
4. The figures and tables are not properly referenced in the text.
5. The methods described under abstract does not adequately described the work done. The statements should be recast.

Experimental design

1. References were not cited for some of the methods used in the experimentation.
2. There are several instances where assertions were not substantiated with references. This has been noted in the appropriate sections of the manuscript.
3. There are several sections that are unnecessarily included under methods (section 2.2,2.7, 2.6).
4. The results should be preferably separated from discussion.

Validity of the findings

1. I’m concerned about the quality of the discussion. The work was poorly discussed. Also, comparison with results of others was poorly done. This is a major concern because as it is currently presented, the discussion is not interesting.
2. a suggestion has been made on how to improve the conclusion.

Additional comments

The authors investigated the efficiency of a newly isolated and characterised bacteria in the bioremediation of industrial effluents containing crystal violet dyes. While the finding is exciting as the isolated bacteria may find useful role in bioremediation, the report was poorly written and sections of the same results repeatedly stated. Some other areas of concerns are highlighted below:

---

## Round 0.2 · accepted · Accept

The authors responded the Reviewer's comments correctly and carefully. The results show potential of bacteria Enterobacter sp. CV–S1 for degradation of organic contaminants such as Crystal Violet. Nice work!